# Dose Intervals and Time since Final Dose on Changes in Metabolic Indices after COVID-19 Vaccination

**DOI:** 10.3390/vaccines11121746

**Published:** 2023-11-23

**Authors:** Amani Alghamdi, Kaiser Wani, Abdullah M. Alnaami, Nasser M. Al-Daghri

**Affiliations:** 1Biochemistry Department, College of Science, King Saud University, Riyadh 11451, Saudi Arabia; 2Chair for Biomarkers of Chronic Diseases, Biochemistry Department, College of Science, King Saud University, Riyadh 11451, Saudi Arabia

**Keywords:** COVID-19 vaccination, dose intervals, metabolic syndrome, metabolic indices, obesity

## Abstract

The rapid development and implementation of COVID-19 vaccines merit understanding its effects on metabolic indices. This retrospective longitudinal study investigated the influence of first-to-second-dose intervals and time since the final dose on the metabolic indices of individuals receiving COVID-19 vaccinations. A total of 318 Saudi subjects (59.7% females) aged 12–60 years received COVID-19 vaccines via the national vaccination program. We collected the anthropometric data and fasting blood samples at specific time points before vaccination and after the final vaccination dose, and biochemical metabolic indices, including glucose and lipid profile, were measured. We also collected the dates of vaccination and COVID-19 history during the study period. The participants were stratified into groups based on first-to-second-dose intervals and time since the final dose to compare pre-and post-vaccination changes in metabolic indices between the groups. Logistic regression analysis revealed no differences in pre- to post-vaccination metabolic status between groups based on first-to-second-dose intervals in either adolescents or adults. However, shorter intervals (≤6 months) between the final dose and follow-up were associated with a decrease in total cardiometabolic components, especially triglyceride levels (OR = 0.39, 95% CI: (0.22–0.68), *p* < 0.001) than longer intervals (>6 months) in adults. In conclusion, time duration since final dose was associated with pre- to post-vaccination changes in metabolic indices, especially triglyceride levels, indicating that post-vaccination improvements wane over time. Further research is needed to validate the observed relationship, as it may contribute to optimizing vaccine effectiveness and safety in the future.

## 1. Introduction

The Coronavirus disease 2019 (COVID-19) pandemic caused by the Severe Acute Respiratory Syndrome Coronavirus 2 (SARS-CoV-2) profoundly affected the global economy, society, and health [1,2]. In response to this crisis, numerous vaccines were developed and deployed in record time to curb the virus’s spread and mitigate the disease’s severity [3,4]. Among them, messenger RNA (mRNA)-based vaccines (such as Pfizer-BioNTech and Moderna) and viral vector vaccines (such as AstraZeneca and Johnson & Johnson) have shown remarkable efficacy in reducing the risk of infection, severe illness, and mortality [5,6,7]. The administration of these vaccines involves a two-dose regimen, with the second dose recommended after an interval that varies across vaccine types and national guidelines [8,9]. For instance, the United States initially recommended a 3-week interval for Pfizer-BioNTech and a 4-week interval for the Moderna vaccines. In contrast, the United Kingdom extended the interval to 12 weeks for both vaccines to prioritize initial doses [10]. Saudi Arabia adopted a 3-week interval for Pfizer-BioNTech and 4 weeks for AstraZeneca, which was later changed to ≤8 weeks [11], while India recommended 4–6 weeks for CoviShield (AstraZeneca) and 28 days for Covaxin [12]. These variations reflect diverse strategies to balance swift immunization with long-term protection.

The Kingdom of Saudi Arabia (KSA) responded swiftly to the COVID-19 pandemic with a comprehensive vaccination program to protect its residents and curb its spread [13,14]. Advised by the Saudi Ministry of Health (MOH) and international health organizations, the government launched its vaccination campaign in December 2020, adopting a phased approach that focused initially on high-risk groups and healthcare workers. In August 2021, the government opened vaccination eligibility to all adults aged 18 years and above [15]. The vaccination program involved procuring and distributing various COVID-19 vaccines as they received regulatory approval, including those developed by Pfizer-BioNTech, AstraZeneca, Moderna, and others [16]. Saudi Arabia’s program has played a crucial role in controlling the disease by significantly reducing the spread of the virus, reducing the severity of cases, and decreasing hospitalizations and deaths, thereby curbing the transmission dynamics of COVID-19 [17].

While the primary goal of COVID-19 vaccination was to prevent viral infection and its associated complications, interest has emerged in understanding the potential impact of vaccination—particularly the effects of different inter-dose intervals—on other health outcomes, including metabolic parameters [18,19,20]. Metabolic disorders, including obesity, type 2 diabetes, and cardiovascular diseases, are associated with increased vulnerability to severe COVID-19 outcomes, so understanding the potential impact of vaccination on metabolic health could have implications for disease prevention and management [21,22,23]. As the global population continues to face the dual challenges of the pandemic and the growing burden of metabolic disorders, integrating metabolic considerations into vaccination strategies can pave the way to improved health outcomes on multiple fronts.

Given the established connections between immune responses and metabolic pathways [18,24], it is plausible that vaccination may trigger metabolic changes. Some reports investigated the effect of improvement in metabolic health like weight loss and blood glucose reduction, etc., in improving the adaptive immune response induced by the COVID-19 vaccines [25,26]. Still, the specific effects of these changes in COVID-19 vaccination and its inter-dose intervals remain poorly understood. This study addressed this research gap by conducting a longitudinal investigation into the effects of inter-dose intervals and time since the final dose on metabolic parameters, including fasting glucose levels, lipid profile, body mass index (BMI), waist circumference, and blood pressure, after COVID-19 vaccination.

## 2. Materials and Methods

### 2.1. Participants and Assessment at Pre-Vaccination Visit

This retrospective longitudinal study continued an educational interventional program conducted by the Chair for Biomarkers in Chronic Diseases (CBCD), King Saud University, in collaboration with the Saudi Charitable Association of Diabetes [27,28,29,30,31]. The study utilized a cluster-randomized sampling approach for selecting 60 high and secondary schools in Riyadh city to conduct an educational program among students and teachers on the rising prevalence of metabolic disorders in Saudi Arabia, including obesity, diabetes, and metabolic syndrome (MetS). The counselling and health educational program included promotional initiatives on balanced dietary habits and curbing sedentary lifestyles through educational lectures and the distribution of educational materials, such as booklets, infographics, and videos, which were delivered physically before the emergence of COVID-19 and via online platforms during the COVID-19–related restrictions. Recruiting for the present study started in November 2020, after COVID-19 restrictions were lifted and before the drive for COVID-19 vaccination began. The pre-vaccination recruitment lasted for five months, during which 318 Saudi adolescents and adults completed the initial assessment, including sociodemographic and anthropometric information and an eight-hour fasting blood sample. There were no specific inclusion criteria, but those with chronic diseases were excluded from the study. The study was approved by the Institutional Review Board (IRB) of the College of Medicine, KSU, Riyadh, Saudi Arabia (no. E-23-7494).

### 2.2. Post-Vaccination Assessment

The recruitment for the post-vaccination visit started in November 2021, with a mean follow-up of 14.08 ± 3.6 months for adolescents and 13.29 ± 3.0 months for adults. At the follow-up visit, apart from the routine eight-hour fasting blood sample and anthropometric assessments, the participants were asked about the COVID-19 vaccination, including information on the type of first, second, and booster doses administered and vaccination dates. They were also asked whether and when they contracted a COVID-19 infection during the study period. The information on dates, vaccine types, and COVID-19 infection was cross-checked with the vaccination and infection records maintained by the MOH, KSA. The study participants recruited for pre- and post-vaccination visits and their COVID vaccination timeline are plotted as a time-series graph and flow chart in Figure 1.

### 2.3. Biochemical Analysis

Fasting blood samples were collected from each participant at pre-vaccination and post-vaccination visits. These were then processed, aliquoted, and transported to the CBCD laboratory for thorough biochemical evaluations. A chemical analyzer (Konelab 20XT, Thermo Scientific, Vantaa, Finland) was used to measure fasting glucose and the lipid profile, which included total cholesterol, HDL-Cholesterol, and triglycerides utilizing commercially available bioassay kits (reference# 981379, 981812, 981823, and 981301, respectively). The total serum 25-hydroxy-Vitamin D (25(OH)D) level was determined using commercial electrochemiluminescence by immunoassays kits from Roche Diagnostics (Indianapolis, IN, USA). The coefficients of intra- and inter-assay differences were 4.6% and 5.3%, respectively. At both visits, glycated hemoglobin (HbA1c) levels were measured using the D-10 Hemoglobin testing equipment (Bio-Rad Laboratories, Hercules, CA, USA), catalog # 220-0201, which utilizes an ion-exchange high-performance liquid chromatography method. 

### 2.4. Data Analysis

The data were analyzed using SPSS version 28.0 (SPSS, Inc., Chicago, IL, USA). To confirm that our data were normally distributed, we utilized the Kolmogorov–Smirnov test. The mean and standard deviation of normally distributed data were reported. Non-normal variables were shown as the median (first, third quartiles). The frequencies (percentages) of categorical variables were shown. The data were separated in both adolescents and adults by first-to-second-dose intervals (≤8 weeks and >8 weeks) and by final dose to post-vaccination visit intervals (≤6 months and >6 months). The final dose was considered as either the second dose (in those participants who didn’t had booster doses) or the booster dose and differences in pre- to post-vaccination changes in metabolic indices were calculated. The pre- to post- changes were checked through paired samples t-test and the mean difference changes in the study groups were compared using independent students t-test. The prevalence of MetS and its components was calculated and presented as *n* (%). A logistic regression analysis was conducted with the first-to-second-dose interval and the final dose to post-vaccination visit interval as the dependent variable and the metabolic indices as independent variables and the ORs were calculated for changes (pre-vaccination to post-vaccination visit 1). The ORs were adjusted for sex, vaccine types at first and second doses, and whether or not they were infected with COVID-19. *p* < 0.05 was considered statistically significant. All of the figures in this research were created using MS Excel 2010.

## 3. Results

### 3.1. Study Subjects’ Baseline Characteristics

The study recruited 318 Saudi subjects, 132 (41.5%) adolescents aged 12–17 years and 186 (58.5%) adults. A total of 59 (44.7%) of the adolescents (Table 1) and 131 (70.4%) of the adults (Table 2) were females. All the subjects had taken their two COVID-19 vaccination doses before follow-up, and most took the Pfizer-BioNTech vaccine as their first (92.8%) and second doses (91.8%), similar in both adolescents and adults. Those who had taken their booster dose before follow-up accounted for 26.5% (*n* = 35) and 49.5% (*n* = 92) of adolescents and adults, respectively. A total of 10 adolescents (7.6%) and 21 adults (14.4%) reported contracting a COVID-19 infection during the study period. All the infected adolescents reported being infected between dose 1 and dose 2, but the adults were infected in equal distribution before dose 1, between dose 1 and 2, and after dose 2. 

Subjects were divided based on dose 1 to dose 2 intervals; those with an interval of 8 weeks or less constituted one group, and those with an interval greater than 8 weeks included the other. The participants were also grouped based on the interval between the final COVID-19 vaccine dose and the follow-up: ≤6 months in one group and >6 months in the other. Table 1 and Table 2 show the groups’ baseline characteristics for adolescents and adults, respectively. No statistical difference was found between the groups in the proportions of vaccine types at dose 1, dose 2, or booster doses, or in the proportions of COVID-19 infections in either adolescents or adults. Furthermore, in both adolescents and adults, the groups were statistically similar in clinical characteristics, such as age, BMI, and male/female ratio.

### 3.2. Differences in Pre- to Post-Vaccination Changes in the Study Groups

Changes in clinical and metabolic parameters pre- and post-vaccination were compared and the differences between groups for adolescents and adults are presented in Table 3. In adolescents, a significant overall increase in weight, BMI, waist and hip circumference, systolic and diastolic pressures, and circulating triglycerides was observed, along with a significant overall increase in HDL cholesterol and vitamin D level post-vaccination. A contrasting overall change was seen in the adults after vaccination, with a significant overall increase in total cholesterol and fasting glucose levels, a significant overall reduction in waist and hip circumferences and systolic and diastolic blood pressure, and a significant overall increase in HDL cholesterol levels in the study groups. When the pre- to post-vaccination difference was compared between the groups based on the first-to-second-dose interval in adolescents and adults, neither the clinical nor the metabolic variables showed any statistically significant difference. When the pre- to post-vaccination differences were compared according to the final dose to follow-up intervals of ≤6 versus >6 months, however, an overall statistically significant reduction in systolic (−10.24 [−17.0, −3.4], *p* = 0.003)) and diastolic blood pressure (−7.90 [−13.4, −2.4], *p* = 0.004) was observed in adolescents, and a significant overall reduction in circulating triglyceride levels (−0.35 [−0.6, −0.1], *p* = 0.003) was observed in adults.

### 3.3. Difference in Pre- to Post-Vaccination Changes in the Prevalence of MetS and Its Components

The study groups’ changes in pre- and post-vaccination in the prevalence of MetS and its components are presented in Table 4. In adolescents, a significant overall increase in the prevalence of central obesity, hypertriglyceridemia, and total number of MetS components was accompanied by an overall decrease in the prevalence of low HDL cholesterol. Adults similarly experienced an overall decrease in the prevalence of low HDL cholesterol from their pre-vaccination to post-vaccination visits, accompanied by a significant decrease in the prevalence of hypertension, MetS, and total number of MetS components (all *p*-values < 0.01). The same trend was seen in the individual study groups, but a logistic regression analysis to compare group differences in these changes found no statistical significance in either adolescents or adults when the subjects were divided based on first-to-second-dose interval periods. When the subjects were divided based on time since the final dose, however, significantly lower odds of high circulating triglyceride levels (OR = 0.39; 95% CI: 0.22–0.68, *p* < 0.001) and total number of MetS components (OR = 0.60: 95% CI: 0.43–0.82, *p* = 0.001) were observed in adults only in the ≤6 months group compared to the >6 months group, even after adjusting for sex, vaccine types at first and second doses, and whether or not they were infected with COVID-19.

## 4. Discussion

This study investigated the effects of different inter-dose intervals on pre- and post-vaccination changes in cardiometabolic indices following COVID-19 vaccination and observed no statistical differences between groups based on first-to-second-dose intervals. To the authors’ knowledge, this study was the first to investigate pre- to post-vaccination changes in metabolic indices following COVID-19 vaccination in Saudi Arabia. When the subjects were categorized based on the recommended first-to-second-dose interval of ≤8 weeks versus >8 weeks, comparable pre- to post-vaccination changes in metabolic indices and number of MetS components were seen between the group in the data of both adolescents and adults, suggesting that there was no profound effect of dose interval on changes in metabolic indices. However, when categorized into groups based on final dose to follow-up interval, a statistically significant decrease in the total number of MetS components in both adolescents and adults, as well as a profound decrease in levels of circulating triglycerides in adults, were seen in the ≤6 months group compared to the >6 months group (Table 4), suggesting that vaccine-induced improvement in metabolic indices wanes over time after the last dose of vaccine.

The COVID-19 pandemic sparked a worldwide immunization campaign to limit viral transmission and severe sickness [32,33]. As the campaign progressed, researchers investigated the effects of COVID-19 vaccination on various specific health aspects, such as infection incidence, hospitalization, rate of recovery, severity, and mortality [6,34,35,36], but few studies examined the nonspecific health effects of COVID-19 vaccination [37,38,39], one of which is changes in metabolic indices after vaccination. For example, a 2021 study by Rubino et al. found that diabetic individuals who received the Pfizer-BioNTech or Moderna COVID-19 vaccines experienced transient increases, typically small, in blood glucose levels after vaccination [40]. The transient hyperglycemia after COVID-19 vaccination may be explained by vaccine-induced immunological reactions and the production of anti-insulin hormones, such as cortisol, catecholamines, etc. [41].

In contrast, adolescents with type 1 diabetes who were vaccinated with either Pfizer BioNTech (BNT162b2) or Moderna (mRNA-1273) COVID-19 vaccines did not experience a change in glycemic control [42]. Our results align with these reports, as changes in fasting glucose levels and the prevalence of hyperglycemia were not significant post-vaccination in either adolescents or adults. One explanation may be that the hyperglycemia found after COVID-19 vaccination in previous studies was short term and lasted only a few days; in our investigation, however, the median follow-up interval was over 12 months. 

Some reports have found that the incidence of dyslipidemia and atherosclerotic cardiovascular disease increases after acute COVID-19 infection and for an indeterminate time afterwards [43,44]. Although the effects of chronic inflammation on lipoprotein metabolism have been widely studied [45,46], few studies have looked at changes in lipid indices after COVID-19 vaccination. A 2021 study by Ramasamy et al. [47] examined the effects of the AstraZeneca vaccine on metabolic parameters and found transient spikes in total and LDL cholesterol levels following vaccination. In the current study, post-vaccination changes in lipids, especially HDL cholesterol, were seen in adolescents and adults. These changes may be connected to the body’s immunological reaction to the vaccine, as immune activity affects lipid metabolism [48,49]. Similarly, there are only a few reported incidents of hypertension after vaccination with COVID-19 mRNA vaccines [50,51,52], and most of those studies are case studies. Variables such as pre-existing conditions, age, and genetics may impact how a person’s metabolic indices react to vaccination. 

Over the past couple of years, a significant discourse has addressed the dosing intervals of the COVID-19 vaccine. Understanding the prospective ramifications of COVID-19 vaccination dosage intervals on the vaccine’s efficacy and nonspecific effects, such as metabolic indices (including blood glucose levels and lipid profiles), is essential for holistic healthcare administration. Initially, the recommended interval between doses for vaccines such as Pfizer and Moderna was 3 weeks [53]. However, due to supply and logistics factors, some countries chose to extend the interval to several weeks or even months, with a recommended ≤8-week interval [54,55], which raised concerns regarding the potential impact of an extended interval between vaccine doses on the vaccine’s efficacy [56]. However, few studies have been conducted to evaluate the influence of the dose intervals on metabolic indices. In the present study, the investigators observed no statistical difference in pre- to post-vaccination changes in the metabolic indices between the groups based on the first-to-second-dose intervals in either adolescents or adults (Table 3 and Table 4). Although the authors could find no relevant studies, a 2021 publication by Steenblock et al. [57] reports the impact of extending the interval between doses of the Pfizer-BioNTech vaccine to 12 weeks, finding no significant effect on the antibody response, which may also explain the observed nonsignificant effect on metabolic indices in the present study. In contrast, some studies have suggested that a delay of at least 9 weeks significantly reduces the risk of infection and death compared to shorter dose intervals [58].

In this study, the regression analysis conducted to explore the differences in pre- to post-vaccination changes in prevalence of MetS and its components (Table 4) was adjusted with whether or not the subject had a prior COVID-19 infection. COVID-19 infection may lead to metabolic dysregulation, possibly impacting post-vaccination outcomes, according to emerging research [23]. The virus has the ability to cause systemic inflammation and disrupt glucose metabolism, resulting in insulin resistance and metabolic dysfunction [59]. Wu et al. [60] found a link between COVID-19 and metabolic abnormalities, emphasizing the need for further research into how past infection may impact the metabolic response to vaccination. Ongoing studies, such as those looking at the post-acute sequelae of SARS-CoV-2 infection (PASC), may offer insight on the complex interaction between viral infections, metabolic changes, and vaccine-induced immune responses [18,61]. Understanding these interactions is critical for determining COVID-19’s long-term effects and improving immunization tactics.

The pre- to post-vaccination changes in metabolic indices observed in this study revealed a more significant reduction in total number of metabolic components in all participants in the group, with an interval of ≤6 months between the last dose and follow-up than the group with an interval of over >6 months. This improvement was especially prevalent in circulating triglycerides in adults, suggesting that the metabolic benefits induced by vaccinations decrease over time. One of the reasons that this effect was seen in adults and not in adolescents in this study may be because adults, generally, tend to have a more mature and experienced immune system compared to adolescents [62]. Over time, exposure to various pathogens builds immunity through the development of memory cells [63]. Although no relevant studies were found that investigated the effect of time interval since completion of COVID-19 vaccination on changes in metabolic indices, COVID-19 immunogenicity data suggest that a decrease in the levels of antibodies over time after receiving two doses of the vaccine may explain these findings [64,65]. It is well known that immune system modulation can significantly affect metabolic indices due to the interconnected nature of the immune system and metabolism [66,67,68]. Interleukin-6 (IL-6) and tumor necrosis factor-alpha (TNF-α), inflammatory molecules released during immune responses, can induce insulin resistance in peripheral tissues, particularly adipose tissue and the liver [69]. Similarly, inflammatory cytokines can influence lipid metabolism by promoting the release of fatty acids from adipose tissue and by increasing the production of very low-density lipoproteins in the liver, which may result in dyslipidemia [70].

The reduction in triglyceride levels after COVID-19 vaccination and the waning effect over time may also be explained by the resolution phase of inflammation [71]. Vaccination aids in the restoration of the immune system to homeostasis by preparing it to react effectively to possible threats [72]. Lipid homeostasis, which contributes to maintaining proper amounts of lipids (including triglycerides and cholesterol) in the circulation, is an essential aspect of overall homeostasis [73]. As part of the body’s general homeostatic functions, triglyceride levels are closely controlled [74]. Another explanation may involve the modulation of the gut microbiota after COVID-19 vaccination, as it plays a critical role in the interplay between the immune system and metabolism [75,76,77].

The decline in lipid index improvements induced by vaccines subsequent to the final dose of COVID-19 vaccination highlights the criticality for continued research and clinical vigilance. Additionally, the results indicate the presence of a phase of recovery mediated by the immune system, specifically in lipid homeostasis, following COVID-19 vaccination. It is essential to comprehend the temporal dynamics of metabolic changes that occur after vaccination in order to maximize the long-term efficacy of vaccines. From a clinical standpoint, these results underscore the significance of customized vaccination approaches, particularly for those who experience a prolonged period of recovery after vaccination. When monitoring patients, clinicians ought to consider the temporal dimension of metabolic alterations and subsequently modify vaccination strategies. This understanding has the potential to provide valuable information for public health campaigns and aid in the improvement of vaccination procedures, thereby ensuring long-lasting metabolic benefits and comprehensive health safeguards. The aforementioned ramifications highlight the fluidity of vaccine-induced impacts and necessitate a sophisticated methodology in scientific investigations and medical environments to maximize the long-term advantages of COVID-19 vaccination on metabolic indicators. Subsequent investigations ought to examine approaches to maintain and augment these outcomes in the long run, possibly by means of targeted interventions or supplemental dosages.

The authors acknowledge some limitations. The researchers did not record specific dietary and physical activity changes at recruitment and after vaccination, which could have influenced the results. The research participants were recruited from schools in the Riyadh region as a continuation of a lifestyle change counselling program. They may not represent the overall population in Saudi Arabia or elsewhere. The national vaccination program-controlled vaccination schedules had shorter intervals between the first and second doses for adolescents than adults, resulting in a disproportionate sample size for adolescents in the study groups, which may have affected the overall results. Among the COVID-19 vaccines, Pfizer-BioNTech was used the most for doses 1 and 2 in our subjects, so inter-vaccine comparison was impossible in this study. Furthermore, this study did not record other factors, such as environmental, social, pre-existing medical conditions and genetic predispositions which may influence metabolic indices and, hence, the resulting observations. Comprehensive research considering these factors may be required to understand the broader impact of vaccination on metabolic health.

## 5. Conclusions

This study suggests that first-to-second-dose interval did not regulate pre- to post-vaccination changes in metabolic indices following COVID-19 vaccination. However, time since COVID-19 vaccination affects changes in metabolic indices, as a statistically significant decrease in the total number of MetS components, with a profound decrease in levels of circulating triglycerides in adults, was seen in the group with ≤6 months between the final vaccine dose and follow-up compared to the >6 months group, suggesting that vaccine-induced improvement in metabolic indices wanes over time after the last dose of vaccine. These observations support the notion of an immunity-mediated recovery phase of homeostasis, especially lipid homeostasis, post-COVID vaccination. However, more studies, especially longitudinal with multi-point follow-up post the last vaccination dose, are needed to corroborate the observed associations, which could contribute to improving vaccination efficacy and safety in the future.

## Figures and Tables

**Figure 1 vaccines-11-01746-f001:**
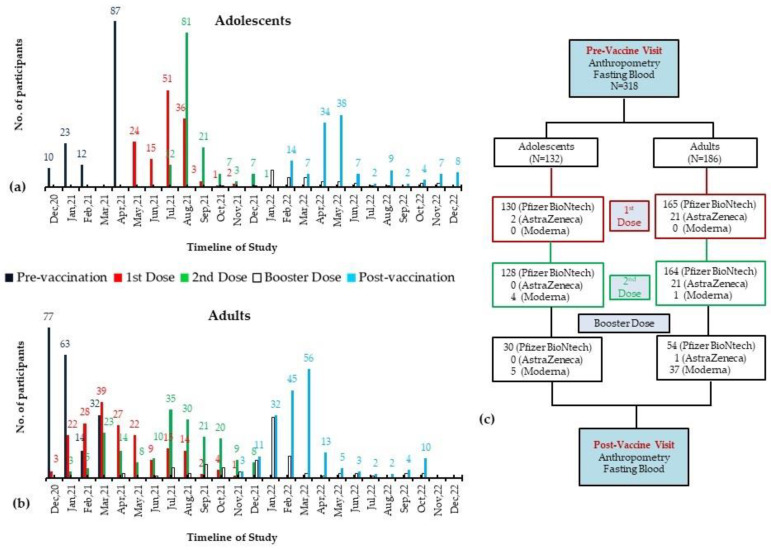
A time-series graph depicting the number of participants recruited during the study period along with their COVID-19 vaccination timeline for adolescents (**a**), adults (**b**), and the flowchart of the study (**c**).

**Figure 2 vaccines-11-01746-f002:**
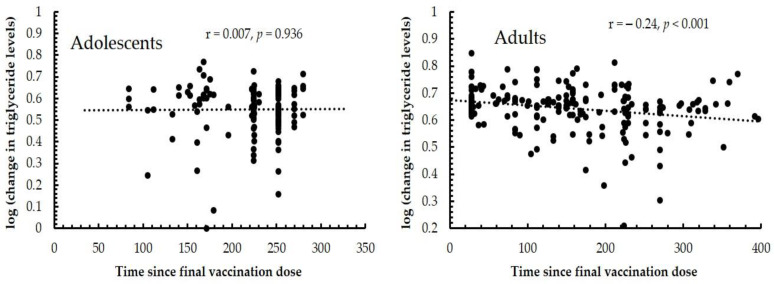
A scatterplot depicting the association between time since final vaccination dose in days against the pre- to post- vaccination changes in triglyceride levels in adolescents and adults. The pre- to post- vaccination changes were transformed to positive numbers by adding 1 + highest increase and then log transformed.

**Table 1 vaccines-11-01746-t001:** COVID-19 vaccination data, clinical and biochemical characteristics of adolescent participants.

	All (*n* = 318)	Adolescents (132)	1st to 2nd Dose Interval			Time since the Final Dose		
			≤8 Weeks (95)	>8 Weeks (37)	*p*-Value	≤6 Months (36)	>6 Months (96)	*p*-Value
1st to 2nd dose (months)	2.89 ± 2.5	2.23 ± 2.1	1.13 ± 0.4	5.05 ± 2.2	<0.001	2.39 ± 2.6	2.17 ± 1.9	0.60
Time to final dose (months)	6.04 ± 2.9	7.4 ± 1.8	7.4 ± 1.8	7.5 ± 1.8	0.63	4.8 ± 0.9	8.4 ± 0.8	<0.001
COVID 19 (1st dose)								
Pfizer-BioNtech	295 (92.8)	130 (98.5)	95 (0.0)	35 (94.6)	0.02	36 (100.0)	94 (97.9)	0.38
AstraZeneca	23 (7.2)	2 (1.5)	0 (0.0)	2 (5.4)		0 (0.0)	2 (2.1)	
Moderna	0 (0.0)	0 (0.0)	0 (0.0)	0 (0.0)		0 (0.0)	0 (0.0)	
COVID 19 (2nd dose)								
Pfizer-BioNtech	292 (91.8)	128 (97.0)	92 (96.8)	36 (97.3)	0.89	36 (100.0)	92 (95.8)	0.21
AstraZeneca	21 (6.6)	0 (0.0)	0 (0.0)	0 (0.0)		0 (0.0)	0 (0.0)	
Moderna	5 (1.6)	4 (3.0)	3 (3.2)	1 (2.7)		0 (0.0)	4 (4.2)	
COVID-19 (Booster dose)								
Pfizer-BioNtech	84 (26.4)	30 (22.7)	21 (22.1)	9 (24.3)	0.16	14 (38.9)	16 (16.7)	0.26
AstraZeneca	1 (0.3)	0 (0.0)	0 (0.0)	0 (0.0)		0 (0.0)	0 (0.0)	
Moderna	42 (13.2)	5 (4.5)	5 (5.3)	0 (0.0)		1 (2.8)	4 (4.2)	
Infected with COVID-19								
*n*	31 (9.7)	10 (7.6)	5 (5.3)	5 (13.5)	0.11	3 (8.3)	7 (7.3)	0.84
After both doses ^a^	7 (22.6)	0 (0.0)	0 (0.0)	0 (0.0)	-	0 (0.0)	0 (0.0)	-
Between doses ^a^	17 (54.8)	10 (100.0)	5 (100.0)	5 (100.0)		3 (100.0)	7 (100.0)	
Before dose 1 ^a^	7 (22.6)	0 (0.0)	0 (0.0)	0 (0.0)		0 (0.0)	0 (0.0)	
Clinical characteristics								
Sex (Females)	190 (59.7)	59 (44.7)	43 (45.3)	16 (43.2)	0.83	17 (47.2)	42 (43.8)	0.72
Age (Years)	28 ± 13.6	15.0 ± 1.2	15.0 ± 1.2	15.1 ± 1.0	0.59	15.2 ± 1.3	14.9 ± 1.1	0.29
Weight (Kg)	65.0 ± 17.9	55.4 ± 17.2	55.3 ± 17.4	55.7 ± 16.8	0.89	59.2 ± 16	53.9 ± 17.4	0.12
BMI (Kg/m^2^)	25.5 ± 6.1	22.7 ± 6.4	22.4 ± 6.5	23.4 ± 6.1	0.43	23.7 ± 6.2	22.3 ± 6.5	0.27
Overweight or obese	152 (47.8)	37 (28.1)	24 (25.3)	13 (35.1)	0.23	11 (30.5)	26 (27.1)	0.87
Waist (cm)	80.1 ± 13.8	74.2 ± 12.7	74.4 ± 12.2	73.7 ± 14.1	0.76	74.2 ± 10.7	74.2 ± 13.4	0.99
Hips (cm)	96.5 ± 14.9	87.79 ± 14.2	87.9 ± 13.8	87.6 ± 15.3	0.91	90.7 ± 11.1	86.7 ± 15.1	0.16
Systolic BP (mmHG)	119.4 ± 18	115.8 ± 17	115.8 ± 18	115.6 ± 16	0.95	122 ± 15.1	113.5 ± 18	0.01
Diastolic BP (mm HG)	73.3 ± 13.2	70.12 ± 14.2	69.8 ± 15.7	70.8 ± 9.5	0.73	73.6 ± 17.5	68.8 ± 12.7	0.09
Biochemical characteristics								
Total cholesterol (mmol/L)	4.8 ± 1	4.3 ± 0.8	4.4 ± 0.7	4.3 ± 0.9	0.52	4.2 ± 0.6	4.4 ± 0.8	0.17
Fasting glucose (mmol/L)	5.4 ± 1.2	5.4 ± 1.1	5.4 ± 1.2	5.3 ± 0.9	0.75	5.3 ± 0.9	5.4 ± 1.2	0.69
HbA1c	5.2 ± 0.7	5.1 ± 0.7	5.1 ± 0.7	5.2 ± 0.6	0.96	5.2 ± 0.5	5.1 ± 0.7	0.64
HDL-Cholesterol (mmol/L)	1.0 ± 0.2	1 (0.9,1.1)	0.99 ± 0.2	1.0 ± 0.2	0.17	0.99 ± 0.2	1.01 ± 0.2	0.50
Triglycerides (mmol/L)	1.1 (0.8,1.5)	0.95 (0.7,1)	0.9(0.7,1)	0.98 (0.7,1)	0.67	0.9 (0.7,1)	0.95 (0.7,1)	0.68
25(OH) D (nmol/L)	30.6 (23.6,44)	29.5 (22.9,37)	30 (23,39)	28.4 (23,32)	0.34	29.8 (22,35)	29 (23,38)	0.73

Note: The data are presented as mean ± standard deviation, median (quartile 1, quartile 3), and *n* (%) for continuous normal, continuous non-normal, and categorical variables, respectively. The superscript a represents variables whose percentages have been calculated from those infected with COVID-19. Differences between the groups were calculated using Student’s *t*-test, Mann–Whitney U test, and the Chi-square test for continuous normal, continuous non-normal, and categorical variables, respectively. *p* < 0.05 was considered statistically significant.

**Table 2 vaccines-11-01746-t002:** COVID-19 vaccination data, clinical and biochemical characteristics of adult participants.

	All Subjects (*n* = 318)	Adults (186)	1st to 2nd Dose Interval			Time since the Final Dose		
			≤8 Weeks (70)	>8 Weeks (116)	*p*-Value	≤6 Months (118)	>6 Months (68)	*p*-Value
1st to 2nd dose (months)	2.9 ± 2.5	3.36 ± 2.5	1.06 ± 0.5	4.75 ± 2.3	<0.001	3.62 ± 2.7	2.91 ± 2.2	0.07
Time to final dose (months)	6.04 ± 2.9	5.07 ± 3.1	5.4 ± 3.5	4.87 ± 2.8	0.26	3.11 ± 1.7	8.47 ± 1.7	<0.001
COVID 19 (1st dose)								
Pfizer-BioNtech	295 (92.8)	165 (88.7)	63 (90.0)	102 (87.9)	0.67	104 (88.1)	61 (89.7)	0.74
AstraZeneca	23 (7.2)	21 (11.3)	7 (10.0)	14 (12.1)		14 (11.9)	7 (10.3)	
Moderna	0 (0.0)	0 (0.0)	0 (0.0)	0 (0.0)		0 (0.0)	0 (0.0)	
COVID 19 (2nd dose)								
Pfizer-BioNtech	292 (91.8)	164 (88.2)	63 (90.0)	101 (87.1)	0.67	104 (88.1)	60 (88.2)	0.74
AstraZeneca	21 (6.6)	21 (11.3)	7 (10.0)	14 (12.1)		13 (11.0)	8 (11.8)	
Moderna	5 (1.6)	1 (0.5)	0 (0.0)	1 (0.9)		1 (0.8)	0 (0.0)	
COVID-19 (Booster dose)								
Pfizer-BioNtech	84 (26.4)	54 (29.0)	23 (32.9)	31 (26.7)	0.56	41 (34.7)	13 (19.1)	0.85
AstraZeneca	1 (0.3)	1 (0.5)	0 (0.0)	1 (0.9)		1 (0.8)	0 (0.0)	
Moderna	42 (13.2)	37 (19.9)	13 (18.6)	24 (20.7)		28 (23.7)	9 (13.2)	
Infected with COVID-19								
*n*	31 (9.7)	21 (14.4)	7 (14.3)	14 (14.4)	0.98	14 (13.5)	7 (16.7)	0.62
After both doses ^a^	7 (22.6)	7 (33.3)	3 (42.9)	4 (28.6)	0.42	6 (42.9)	1 (14.3)	0.42
Between doses ^a^	17 (54.8)	7 (33.3)	1 (14.3)	6 (42.9)		4 (28.6)	3 (42.9)	
Before dose 1 ^a^	7 (22.6)	7 (33.3)	3 (42.9)	4 (28.6)		4 (28.6)	3 (42.9)	
Clinical characteristics								
Sex (Females)	190 (59.7)	131 (70.4)	48 (68.6)	83 (71.6)	0.67	84 (71.2)	47 (69.1)	0.77
Age (Years)	28 ± 13.6	37.23 ± 10.5	36.77 ± 10.6	37.51 ± 10.4	0.64	38.08 ± 10	35.76 ± 11.2	0.15
Weight (Kg)	65.0 ± 17.9	71.8 ± 15	71.9 ± 16.8	71.8 ± 13.9	0.98	72 ± 14.4	71.6 ± 16.1	0.87
BMI (Kg/m^2^)	25.5 ± 6.1	27.49 ± 5.1	27.69 ± 5.5	27.37 ± 4.8	0.68	27.52 ± 5	27.45 ± 5.2	0.93
Overweight or obese	152 (47.8)	115 (61.8)	41 (58.6)	74 (63.7)	0.28	72 (61.0)	43 (63.3)	0.34
Waist (cm)	80.1 ± 13.8	84.2 ± 13	84.5 ± 13.7	84.1 ± 12.6	0.85	84.8 ± 12.9	83.3 ± 13.2	0.46
Hips (cm)	96.5 ± 14.9	102.7 ± 12	102 ± 13.1	103 ± 11.4	0.70	103 ± 11.6	101 ± 12.7	0.27
Systolic BP (mmHG)	119 ± 18.1	122 ± 18.3	121 ± 15.2	122 ± 20	0.94	121 ± 16.4	124 ± 21.1	0.17
Diastolic BP (mm HG)	73.1 ± 13.2	75.6 ± 12	74.4 ± 10.6	76.2 ± 12.7	0.32	76 ± 12.9	75 ± 10.1	0.55
Biochemical characteristics								
Total cholesterol (mmol/L)	4.8 ± 1	5.2 ± 1	5.1 ± 1.2	5.2 ± 0.9	0.45	5.1 ± 1	5.2 ± 1.1	0.77
Fasting glucose (mmol/L)	5.4 ± 1.2	5.4 ± 1.2	5.3 ± 1.1	5.5 ± 1.3	0.18	5.5 ± 1.3	5.3 ± 1	0.33
HbA1c	5.2 ± 0.7	5.2 ± 0.8	5.2 ± 0.6	5.2 ± 0.8	0.74	5.2 ± 0.8	5.13 ± 0.7	0.51
HDL-Cholesterol (mmol/L)	1.0 ± 0.2	1 (0.9,1.1)	1.02 ± 0.2	1.0 ± 0.2	0.76	0.99 ± 0.2	1.0 ± 0.2	0.19
Triglycerides (mmol/L)	1.1 (0.8,1.5)	1.2 (0.9,1.7)	1.2 (0.9,2)	1.2 (1,1.9)	0.36	1.2 (0.9,2)	1.18 (1,1.7)	0.83
25(OH) D (nmol/L)	31 (23.6,43.5)	32.7 (24,47)	32 (23,52)	34 (24,45)	0.67	34.4 (24,48)	32 (23,44)	0.40

Note: The data are presented as mean ± standard deviation, median (quartile 1, quartile 3), and *n* (%) for continuous normal, continuous non-normal, and categorical variables, respectively. The superscript a represents variables in which percentages have been calculated from those infected with COVID-19. Differences between the groups were calculated using Student’s *t*-test, Mann–Whitney U test, and Chi-square test for continuous normal, continuous non-normal, and categorical variables, respectively. *p* < 0.05 was considered statistically significant.

**Table 3 vaccines-11-01746-t003:** Differences in pre- and post-vaccination changes in clinical and metabolic parameters among groups based on the first-to-second-dose dose interval and time since the final dose.

	Changes in 1st to 2nd Dose Interval				Changes in the Time since the Final Dose			
	≤8 Weeks	>8 Weeks	Difference Change (95% CI)	*p*	≤6 Months	>6 Months	Difference Change (95% CI)	*p*
Adolescents								
Weight (Kg)	11.5	6.99	4.5 (−1.3,10)	0.13	9.1	10.67	−1.6 (−7.5,4)	0.60
BMI (Kg/m^2^)	4.83	3.15	1.7 (−0.7,4)	0.17	3.78	4.57	−0.8 (−3.2,1.6)	0.52
Waist (cm)	6.24	4.75	1.5 (−4.1,7)	0.60	7.60	5.16	2.4 (−3.2,8.1)	0.39
Hips (cm)	8.57	7.09	1.5 (−3.7,7)	0.57	6.11	8.92	−2.8 (−8.0,2.4)	0.28
Systolic BP (mmHG)	5.18	0.78	4.4 (−2.5,11)	0.21	−3.50	6.74	−10.2 (−17,−3.4)	0.003
Diastolic BP (mm HG)	3.14	3.03	0.1 (−5.5,6)	0.97	−2.64	5.26	−7.9 (−13,−2.4)	0.005
Total Chol (mmol/L)	0.09	−0.08	0.2 (−0.1,0.5)	0.29	0.12	−0.06	0.2 (0.1,0.5)	0.05
Fasting glucose (mmol/L)	−0.02	−0.04	0.02 (−0.3,0.3)	0.92	−0.13	0.00	−0.14 (−0.5,0.2)	0.41
HbA1c	−0.07	−0.11	0.04 (−0.3,0.4)	0.81	−0.24	−0.03	−0.21 (−0.5,0.1)	0.21
HDL-Chol (mmol/L)	0.27	0.13	0.14 (0.0,0.3)	0.11	0.29	0.21	0.09 (−0.1,0.3)	0.30
Triglycerides (mmol/L)	0.75	0.67	−0.11 (−0.4,0.2)	0.48	0.57	0.75	−0.16 (−0.5,0.2)	0.34
25(OH)D (nmol/L)	7.20	10.1	−4.96 (−15.5,5.6)	0.42	10.05	7.47	−2.7 (−13.3,8.0)	0.62
Adults								
Weight (Kg)	2.46	0.96	1.49 (−1.2,4.2)	0.27	1.70	1.22	0.47 (−2.2,3.2)	0.73
BMI (Kg/m^2^)	0.97	0.40	0.57 (−0.4,1.6)	0.27	0.66	0.51	0.15 (−0.9,1.2)	0.77
Waist (cm)	−3.17	−3.94	0.77 (−3.4,4.9)	0.71	−3.57	−3.78	0.22 (−3.9,4.4)	0.92
Hips (cm)	−4.81	−8.28	3.48 (−1.2,8.2)	0.14	−7.42	−6.18	−1.24 (−6.0,3.5)	0.61
Systolic BP (mmHG)	−9.76	−8.03	−1.73 (−7.9,4.5)	0.58	−7.53	−10.67	3.13 (−3.1,9.3)	0.32
Diastolic BP (mm HG)	−2.64	−2.76	0.12 (−3.8,4.1)	0.95	−3.4	−1.53	−1.87 (−5.8,2.1)	0.35
Total Chol (mmol/L)	0.33	0.28	0.04 (−0.3,0.4)	0.79	0.36	0.22	0.13 (−0.2,0.5)	0.40
Fasting glucose (mmol/L)	−0.13	−0.07	−0.06 (−0.7,0.5)	0.84	−0.21	0.12	−0.33 (−0.9,0.3)	0.28
HbA1c	0.20	0.36	−0.16 (−0.5,0.2)	0.35	0.35	0.21	0.13 (−0.2,0.5)	0.44
HDL-Chol (mmol/L)	0.38	0.42	−0.04 (−0.2,0.1)	0.60	0.45	0.35	0.1 (0.0,0.2)	0.17
Triglycerides (mmol/L)	−0.10	−0.01	0.01 (−0.2,0.2)	0.95	−0.11	0.14	−0.4 (−0.6,−0.1)	0.003
25(OH)D D (nmol/L)	−4.27	12.48	−9.23 (−24,5.6)	0.21	6.75	0.35	0.03 (−15.6,16)	1.0

Note: The data are presented as mean and median differences (post-vaccination—pre-vaccination) in the groups for continuous normal and continuous non-normal variables, respectively. The data representing differences in changes between the groups are presented by mean difference change and associated 95% confidence interval. The *p*-values for pre- to post-vaccination changes among each group were presented in Appendix A. *p* < 0.05 was considered statistically significant.

**Table 4 vaccines-11-01746-t004:** Differences in pre- to post-vaccination changes in the prevalence of MetS and its components by study groups.

Mets Components	Pre- Vaccine	Post- Vaccine	Differences in % from Pre- to Post-Vaccine Visits					Odds Ratio, 95% Confidence Interval, *p*		
			All	1st to 2nd Dose (Weeks)		Time to Final Dose (Months)				
				≤8	>8	≤6	>6	Parameters	≤8 vs. >8 Weeks	≤6 vs. >6 Months
Adolescents										
Central Obesity	7 (5.3)	24 (18.2)	12.9	13.6	10.9	13.8	12.5	Waist circumference1	1.00, (0.9,1.0), 0.597	1.01, (0.9,1.0), 0.393
Hyperglycemia	12 (9.1)	17 (12.9)	3.8	0.0	13.5	0.0	5.2	Fasting glucose1	1.02, (0.7,1.6), 0.919	0.83, (0.5,1.3), 0.407
Low HDL	82 (62.1)	53 (40.2)	−21.9	−25.3	−13.6	−20.6	−14.5	HDL-Cholesterol1	2.09, (0.8,5.2), 0.113	1.59, (0.7,3.8), 0.294
High Triglycerides	26 (19.7)	85 (64.4)	44.7	47.4	37.9	30.6	50.0	Triglycerides1	0.85, (0.5,0.29), 0.479	0.79, (0.5,1.3), 0.338
Hypertension	30 (22.7)	28 (21.2)	−1.5	−5.3	8.1	−16.7	4.2	Systolic BP1	1.01, (0.9,1.0), 0.213	0.97, (0.94,1.00), 0.056 ^a^
								Diastolic BP1	1.00, (0.9,1.0), 0.969	0.96, (0.93,1.00), 0.062 ^a^
Mets	11 (8.3)	22 (16.7)	8.4	4.2	18.9	0.0	11.4	-	-	-
MetS comp.	1.19 ± 1.0	1.57 ± 1.0	0.38	0.31	0.57	−0.14	0.58	MetS components1	0.83, (0.6,1.2), 0.268	0.63, (0.44,0.89), 0.011 ^a^
Adults										
Central Obesity	46 (24.7)	39 (21.0)	−3.7	1.4	−6.9	−5.9	0.0	Waist circumference1	1.00, (0.9,1.0), 0.712	1.00, (0.9,1.0), 0.918
Hyperglycemia	47 (25.3)	35 (18.8)	−6.5	−7.2	−6.0	−10.2	0.0	Fasting glucose1	0.99, (0.8,1.1), 0.839	0.92, (0.8,1.1), 0.282
Low HDL	147 (79.0)	55 (29.6)	−49.4	−47.1	−50.8	−55.1	−39.7	HDL-Cholesterol1	0.84, (0.4,1.6), 0.601	1.59, (0.8,3.0), 0.168
High Triglycerides	48 (25.8)	41 (22.0)	−3.8	−4.3	−3.5	−11.9	10.3	Triglycerides1	1.01, (0.7,1.5), 0.951	0.39, (0.2,0.7), <0.001 ^a^
Hypertension	54 (29.0)	14 (7.5)	−21.5	−21.4	−21.5	−22.9	−19.2	Systolic BP1	0.99, (0.9,1.0), 0.581	1.01, (0.9,1.0), 0.323
								Diastolic BP1	1.00, (0.9,1.0), 0.953	0.99, (0.9,1.0), 0.351
MetS	48 (25.8)	21 (11.3)	−14.5	−12.8	−15.5	−16.1	−11.8	-	-	-
MetS comp.	1.84 ± 1.2	0.99 ± 1.1	−0.85	−0.79	−0.89	−1.06	−0.49	MetS components1	1.06, (0.9,1.3), 0.627	0.60, (0.43,0.82), 0.001 ^a^

Note: The prevalence of MetS and its components is presented by *n* (%). The pre- and post-vaccination changes are presented by the prevalence percentage from pre-vaccination to post-vaccination visits, calculated for all four study groups. The *p*-values for pre- to post-vaccination changes in MetS components among each group were presented in Appendix A. A logistic regression analysis was conducted with the first-to-second-dose interval and the final vaccination to post-vaccination visit interval as dependent variables; the ORs were calculated for changes (pre-vaccination to post-vaccination visit 1) in the reported independent variables. The superscript a represents significant ORs after adjusting for sex, vaccine types at first and second doses, and whether or not infected with COVID-19. *p* < 0.05 was considered statistically significant. A scatterplot depicting the correlation between time since the final dose and the pre- to post-vaccination changes in triglyceride levels is provided as Figure 2. Another figure depicting the correlation between time since the first dose and the pre- to post-vaccination changes in triglyceride levels is provided as Appendix A.

## Data Availability

The corresponding author may provide the data described in this research upon request.

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
