# Peer review of "Dose Intervals and Time since Final Dose on Changes in Metabolic Indices after COVID-19 Vaccination"

_vaccines, 2023, doi:10.3390/vaccines11121746_

Round 1

Reviewer 1 Report

Comments and Suggestions for Authors

The study is interesting and deals with the metabolic changes induced by the COVID-19 vaccine, differentiating them by type of vaccine. it is quite innovative however there is an important bias. In fact, we know nothing about the pathologies suffered by patients and we know nothing about their medical history. Furthermore, how do we know whether the metabolic changes were induced by the vaccine or the stress of the pandemic? So the authors should introduce a paragraph within the study entitled "Limitation" and include these doubts that I just raised.

They should also cite these articles:

https://doi.org/10.1111/obr.13313

https://doi.org/10.1515/dmpt-2022-0148

https://doi.org/10.3390/healthcare10020319

https://doi.org/10.3390/cells11071241

https://doi.org/10.3390/vaccines10010079

https://doi.org/10.3390/vaccines11010142

https://doi.org/10.1124/dmd.122.000934

Minor revisions

severe acute respiratory syndrome coronavirus 2 should be entered with a capital letter

Author Response

The study is interesting and deals with the metabolic changes induced by the COVID-19 vaccine, differentiating them by type of vaccine. it is quite innovative however there is an important bias. In fact, we know nothing about the pathologies suffered by patients and we know nothing about their medical history. Furthermore, how do we know whether the metabolic changes were induced by the vaccine or the stress of the pandemic? So, the authors should introduce a paragraph within the study entitled "Limitation" and include these doubts that I just raised.

Response: The authors are thankful to the reviewer for these points. We agree with the reviewer that factors like pre-existing diseases and genetic predispositions in the study subjects may have influenced the results. Our study subjects were taken from an apparently healthy cohort from a lifestyle change counselling program in schools. However, we are aware of this limitation raised by the reviewer and this was in fact provided as a limitation in the last paragraph of discussion, which we have modified in the revised draft based on reviewer’s suggestion. The reviewer may also agree that it is very difficult to control all the factors and covariates that may have affected the results; however, the beauty of the longitudinal studies is that the baseline condition act as the reference which helps in observing the changes. We are also aware that a multi-point follow-up after the last vaccination dose may have provided better and clearer results and hence results may be verified with such longitudinal multi-point studies, provided as a suggestion in this study.  

They should also cite these articles:

https://doi.org/10.1111/obr.13313

https://doi.org/10.1515/dmpt-2022-0148

https://doi.org/10.3390/healthcare10020319

https://doi.org/10.3390/cells11071241

https://doi.org/10.3390/vaccines10010079

https://doi.org/10.3390/vaccines11010142

https://doi.org/10.1124/dmd.122.000934

Response: The authors are thankful to the reviewer for suggesting these references. In fact, most of these references are pertinent to the study and have been included in the revised draft.

Minor revisions

severe acute respiratory syndrome coronavirus 2 should be entered with a capital letter

Response: Done, thanks

Reviewer 2 Report

Comments and Suggestions for Authors

Alghamdi et al report data from a retrospective longitudinal study investigating the link between covid 19 vaccine dose intervals and metabolic changes. Although in the introduction it is reported that” Given the established connections between immune responses and metabolic path-70 ways [18,23], it is plausible that vaccination may trigger metabolic changes”, the supporting references only link bidicrectionally the immune and metabolic system. Indeed, plenty of data supports that metabolic changes can affect the immune system even after covid 19 vaccination (10.3390/vaccines10010079), and that metabolic health is a strong predictor of vaccines response overall (10.1016/j.vaccine.2015.06.101). The authors may agree that they could be looking at their data in this sense (metabolic health affecting immune response, not the opposite) and may want to reconsider the interpretation of these data. Such a contribution would be very valuable and strongly supported by scientific literature.

Author Response

Alghamdi et al report data from a retrospective longitudinal study investigating the link between covid 19 vaccine dose intervals and metabolic changes. Although in the introduction it is reported that” Given the established connections between immune responses and metabolic path-70 ways [18,23], it is plausible that vaccination may trigger metabolic changes”, the supporting references only link bidicrectionally the immune and metabolic system. Indeed, plenty of data supports that metabolic changes can affect the immune system even after covid 19 vaccination (10.3390/vaccines10010079), and that metabolic health is a strong predictor of vaccines response overall (10.1016/j.vaccine.2015.06.101). The authors may agree that they could be looking at their data in this sense (metabolic health affecting immune response, not the opposite) and may want to reconsider the interpretation of these data. Such a contribution would be very valuable and strongly supported by scientific literature.

Response: The authors thank the reviewer for this valuable insight into the data. We were aware that some studies were done to show the effect of improvement in metabolic health like weight loss, and blood glucose reduction, etc. in improving the adaptive immune response induced by the COVID-19 vaccines. However, in this study, we tried to investigate the association between interdose intervals and time since the last vaccine dose with improvements in metabolic health. Though the reviewer’s observations are pertinent and may provided valuable results from the Saudi COVID-19 immunization program data which may be a good separate research topic to explore, yet we believe that the observations presented here in this study especially that “post-vaccination improvements wane over time” may still be useful as observations like these support the notion of an immunity-mediated recovery phase of homeostasis, especially lipid homeostasis, post-COVID vaccination. We tried to incorporate the reviewers point in our hypothesis, in the last paragraph of the revised introduction section to give readers some more perspective to explore this topic.

Reviewer 3 Report

Comments and Suggestions for Authors

This study demonstrated the longitudinal impact of SARS-CoV-2 vaccination on metabolic indices. However, several clarifications are necessary to enhance the comprehension and robustness of the findings:

The methodology would benefit from if the same patients were followed at different time points after their final vaccination. Such a longitudinal approach would likely strengthen the conclusions. It may still be informative to plot triglyceride levels against the days following the last vaccination for the adolescent or adult groups to investigate any potential negative correlation. Similarly, it would be interesting to explore how these levels correlate with the days post-first vaccination.

Another point to consider is the inclusion of asymptomatic COVID-19 infections in the discussion, as the possibility of such infections cannot be disregarded.

Regarding the mRNA vaccines, it would be pertinent to mention if similar effects on metabolites have been observed with other types of vaccines used against SARS-CoV-2.

In the hypothesis addressing the decrease in triglycerides in the early post-vaccination stage in adults, it would be useful to theorize why adolescents may not exhibit these effects.

Figure Clarification: Figure 1 presents confusion regarding the study design. A brief illustration of the participants and grouping as well as sample collection should be clearly graphed.

Table: The head of table 2 was wrong. 

Additionally, the study should specify whether recent infections or vaccinations were accounted for before the assessments following the last vaccination.

Lastly, a more comprehensive explanation of the statistical methods applied in this study is necessary for a clearer understanding of the findings.

Comments on the Quality of English Language

Minor editing of English language required

Author Response

This study demonstrated the longitudinal impact of SARS-CoV-2 vaccination on metabolic indices. However, several clarifications are necessary to enhance the comprehension and robustness of the findings:

The methodology would benefit from if the same patients were followed at different time points after their final vaccination. Such a longitudinal approach would likely strengthen the conclusions. It may still be informative to plot triglyceride levels against the days following the last vaccination for the adolescent or adult groups to investigate any potential negative correlation. Similarly, it would be interesting to explore how these levels correlate with the days post-first vaccination.

Response: We agree with the reviewer that a multi-point follow-up post final vaccination dose for the same individuals would have produced stronger results than a single point than in this study. However, as per the reviewer’s suggestion, a scatterplot depicting the association of time since the final dose and pre to post vaccination change in triglyceride levels was done and added in the revised draft which indeed showed a negative correlation in adults. Similarly, a scatterplot with time since first vaccination dose was also done and provided as a supplementary graph. 

Another point to consider is the inclusion of asymptomatic COVID-19 infections in the discussion, as the possibility of such infections cannot be disregarded.

Response: We are thankful to the reviewer for this suggestion. In fact, the participants with a self-reported COVID-19 infection between the follow-ups was reported in table 1 (31 out of total 318). Also, the regression analysis conducted to explore the differences in pre to post vaccination changes in prevalence of MetS and its components was adjusted with this variable of “whether or not the subject had COVID-19 infection”. This was discussed in a newly added paragraph (paragraph 6) of the revised discussion.  

Regarding the mRNA vaccines, it would be pertinent to mention if similar effects on metabolites have been observed with other types of vaccines used against SARS-CoV-2.

Response: More than 90% of our study subjects (in adolescents as well as adults) had taken Pfizer-BioNTech (mRNA vaccine) as their first and second doses. Hence, our results are mostly pertinent to this type of vaccine and inter-vaccine comparison was not possible because of low sample size for other type of vaccine (viral vector). This was mentioned in the limitation section. We however believe that the observations may be extended to other types of vaccines such as viral vector vaccines as the changes in metabolic indices was partly explained by the immunogenicity induced by the vaccines and both these types of vaccines were proven to be effective in their fight against COVID-19. Longitudinal studies such as this one may still be needed to strengthen this point.  

In the hypothesis addressing the decrease in triglycerides in the early post-vaccination stage in adults, it would be useful to theorize why adolescents may not exhibit these effects.

Response: This is a great suggestion. Although no relevant studies were found that investigated the effect of time interval since completion of COVID-19 vaccination on changes in metabolic indices, we believe that the discrepancy in results between adolescent and adult data was probably because adults, generally, tend to have a more mature and experienced immune system compared to adolescents. And over time, exposure to various pathogens builds immunity through the development of memory cells. This has been mentioned in the revised discussion section (para 7).

Figure Clarification: Figure 1 presents confusion regarding the study design. A brief illustration of the participants and grouping as well as sample collection should be clearly graphed.

Response: Figure 1 has been replotted in the revised draft. Also, a flowchart to depict the stages of the study has been added in the figure.

Table: The head of table 2 was wrong. 

Response: Tables 1 and 2 provided the COVID-19 vaccination data, clinical and biochemical characteristics of adolescents and adults respectively. This has been modified as suggested by the reviewer.

Additionally, the study should specify whether recent infections or vaccinations were accounted for before the assessments following the last vaccination.

Response: This has been accounted for in the study. The regression analysis presented in table 4 has been adjusted with whether or not the participant reported COVID-infection. Also, the final dose referred to either the 2nd dose (in those participants who didn’t had booster doses) or the booster dose. This was mentioned in the revised draft.

Lastly, a more comprehensive explanation of the statistical methods applied in this study is necessary for a clearer understanding of the findings.

Response: We thank the reviewer for the suggestion. Two new sections (2.3 and 2.4) for explanation into biochemical and statistical analysis respectively has been added in the revised draft.

Reviewer 4 Report

Comments and Suggestions for Authors

Vaccination against COVID-19 has been a crucial tool in mitigating the spread and severity of the disease. However, it is essential to understand the potential effects of vaccination on metabolic health. Studying the relationship between COVID-19 vaccination and metabolic indices can provide valuable insights into the overall health effects of vaccination and help inform public health strategies. Retrospective longitudinal studies like this one play a crucial role in assessing the real-world effects of COVID-19 vaccines on various health parameters. They help in understanding the long-term implications of vaccination and provide valuable information for public health strategies. The topic is important and the manuscript provides a comprehensive analysis of the subject. I recommend accepting this article after MINOR REVISIONS.

1.     The specific effects of COVID-19 vaccination on metabolic indices may vary among individuals and depend on various factors, including pre-existing metabolic conditions and genetic predispositions. Therefore, comprehensive research considering these factors is necessary to understand the broader impact of vaccination on metabolic health.

2.     These findings provide valuable insights into the potential impact of time duration to the final dose on post-vaccination changes in metabolic indices. The observed decrease in triglyceride levels indicates that post-vaccination improvements may diminish over time. However, it is important to note that further research is needed to validate these findings and understand the underlying mechanisms. This will contribute to optimizing vaccine effectiveness and safety in the future.

3.     Discuss the implications of your findings for future research or clinical practice.

4.     Check the abbreviations throughout the manuscript. For example, in line 32, “COVID-19”--- “coronavirus disease 2019 (COVID-19)”.

Comments on the Quality of English Language

 Minor editing of the language

Author Response

Vaccination against COVID-19 has been a crucial tool in mitigating the spread and severity of the disease. However, it is essential to understand the potential effects of vaccination on metabolic health. Studying the relationship between COVID-19 vaccination and metabolic indices can provide valuable insights into the overall health effects of vaccination and help inform public health strategies. Retrospective longitudinal studies like this one play a crucial role in assessing the real-world effects of COVID-19 vaccines on various health parameters. They help in understanding the long-term implications of vaccination and provide valuable information for public health strategies. The topic is important and the manuscript provides a comprehensive analysis of the subject. I recommend accepting this article after MINOR REVISIONS.

Response: We are thankful to the reviewer and agree that the findings presented in the study may help play an important role in understanding the overall health effects of COVID-19 vaccination.

  1. The specific effects of COVID-19 vaccination on metabolic indices may vary among individuals and depend on various factors, including pre-existing metabolic conditions and genetic predispositions. Therefore, comprehensive research considering these factors is necessary to understand the broader impact of vaccination on metabolic health.

Response: We agree with the reviewer that factors like pre-existing diseases and genetic predispositions in the study subjects may have influenced the results. Our study subjects were taken from an apparently healthy cohort from a lifestyle change counselling program in schools. However, we are aware of this limitation raised by the reviewer and this was in fact provided as a limitation in the last section, which we have modified in the revised draft based on reviewer’s suggestion. 

  1. These findings provide valuable insights into the potential impact of time duration to the final dose on post-vaccination changes in metabolic indices. The observed decrease in triglyceride levels indicates that post-vaccination improvements may diminish over time. However, it is important to note that further research is needed to validate these findings and understand the underlying mechanisms. This will contribute to optimizing vaccine effectiveness and safety in the future.

Response: We appreciate the reviewer for noticing the valuable insights provided on the observations in the study. The waning of improvements in lipid indices post last vaccination dose may explain in part the immunity-mediated recovery phase of lipid homeostasis. As rightly pointed out by the reviewer, the observations presented need to be confirmed with further studies including the longitudinal ones with more than one follow-up post the last vaccination dose. This has been added in the revised conclusion section. 

  1. Discuss the implications of your findings for future research or clinical practice.

Response: We are thankful to the reviewer for this suggestion. A new paragraph before the limitation section has been added discussing the implications of the findings.

  1. Check the abbreviations throughout the manuscript. For example, in line 32, “COVID-19”--- “coronavirus disease 2019 (COVID-19)”.

Response: Abbreviations has been checked throughout the manuscript and the full-form has been provided wherever first mentioned.

Round 2

Reviewer 1 Report

Comments and Suggestions for Authors

The article has improved considerably and is ready for publication

Author Response

We thank the reviewer for appreciating our revisions.

Reviewer 2 Report

Comments and Suggestions for Authors

the authors addressed my concerns

Author Response

(The authors gave the same response as above.)

Reviewer 3 Report

Comments and Suggestions for Authors

The manuscript has been greatly improved. No further revision this round.

Author Response

We appreciate the reviewer's compliments with our revised work.